# Seasonal Patterns and Trends in Dermatoses in Poland

**DOI:** 10.3390/ijerph19158934

**Published:** 2022-07-22

**Authors:** Krzysztof Bartosz Klimiuk, Dawid Krefta, Karol Kołkowski, Karol Flisikowski, Małgorzata Sokołowska-Wojdyło, Łukasz Balwicki

**Affiliations:** 1Faculty of Medicine, Medical University of Gdańsk, 80-210 Gdańsk, Poland; 2Faculty of Electronics, Telecommunications and Informatics, Gdansk University of Technology, 80-233 Gdańsk, Poland; s173355@student.pg.edu.pl; 3Dermatological Students Scientific Association, Department of Dermatology, Venerology and Allergology, Faculty of Medicine, Medical University of Gdańsk, 80-210 Gdańsk, Poland; karolkolkowski@gumed.edu.pl; 4Faculty of Management and Economics, Gdańsk University of Technology, 80-233 Gdańsk, Poland; karol.flisikowski@pg.edu.pl; 5Department of Dermatology, Venereology and Allergology, Medical University of Gdańsk, 80-210 Gdańsk, Poland; malgorzata.sokolowska-wojdylo@gumed.edu.pl; 6Department of Public Health and Social Medicine, Medical University of Gdańsk, 80-210 Gdańsk, Poland; lukasz.balwicki@gumed.edu.pl

**Keywords:** Google Trends, trends, seasonality, dermatoses, STD, infodemiology

## Abstract

Background: The amount of data available online is constantly increasing, including search behavior and tracking trends in domains such as Google. Analyzing the data helps to predict patient needs and epidemiological events more accurately. Our study aimed to identify dermatology-related terms that occur seasonally and any search anomalies during the SARS-CoV-2 pandemic. Methods: The data were gathered using Google Trends, with 69 entries between January-2010 and December-2020 analyzed. We conducted the Seasonal Mann–Kendal Test to determine the strength of trends. The month with the highest seasonal component (RSV) and the lowest seasonal component (RSV) was indicated for every keyword. Groups of keywords occurring together regularly at specific periods of the year were shown. Results: We found that some topics were seasonally searched in winter (e.g., herpes, scabies, candida) and others in summer (e.g., erythema, warts, urticaria). Conclusions: Interestingly, downward trends in searches on sexually transmitted diseases in comparison with increased infection rates reported officially show a strong need for improved sexual education in Poland. There were no significant differences in trends for coronavirus-related cutaneous symptoms during 2020. We have shown that the seasonality of dermatologically related terms searched in Poland via Google did not differ significantly during SARS-CoV-2 pandemic.

## 1. Introduction

Approximately 5 million people with various skin diseases live in Poland [1] and roughly 29% of Poles have ever reported skin conditions [2]. Many patients search for information about their complaints and health advice online [3]. Increasing access to technology, but also awareness of their health, cause people to invest more and more energy into maintaining their health, using the Internet in order to improve their health but also to reduce medical expenses [4]. Another motivation for seeking medical advice online is stigma associated with conditions such as mental illness, so patients seek solutions on their own [5]. It can be assumed that search volumes increased due to the impact of the COVID-19 pandemic restricting access to dermatology professionals [6,7,8], as was the case with flu-like symptoms [8] or with insomnia [9].

Google is the most commonly used search engine in the world [10] and in Poland [11]. The records of search volumes and trends in searches can be used for further analysis of the behavior and searches on the Internet. People’s use of social media and search engines generates vast amounts of data [12], which, apart from monitoring the occurrence of symptoms, can also be used, for example, to track the popularity of smoking cessation [13,14]. The tool provided by Google—Google Trends—enables estimation of the relative search volumes (RSV) for a selected word or phrase by users in a selected region and time period [15]. This tool has been used for research in the fields of medicine for studying, among others, hypertension [16], pain [17], antibiotics [18] or the above-mentioned dermatological diseases [19]. This kind of research complements epidemiological data, as well as providing new insights for a more holistic picture of health [20].

Health information behavior, including seeking, obtaining and using health-related information, is associated with the cyclical nature of health and illness. Understanding of this cyclical process could be used to facilitate patients’ empowerment and promote pro-health behaviors [21]. However, to the best of our knowledge, no study has used the available infodemiology data before to check the seasonality of dermatological disorders in Poland. Such knowledge may help us to better understand patients’ perceptions of skin diseases, which may result in the better preparation of primary care doctors and dermatologists. Therefore, our study aims to identify which dermatological symptoms/diseases are characterized by seasonality in their Google search patterns.

## 2. Materials and Methods

### 2.1. Data Collection

We conducted cross-sectional time series analysis to distinguish seasonal patterns among diseases. Google Trends is a tool that facilitates the tracking of the relative frequency of searches in time in a provided location-relative search volume (RSV). Periods when the given word was searched in a small number compared to the maximum of searches or not at all mean RSV = 0. The peak of popularity defines RSV = 100. Likewise, when a given term was searched with 50% frequency of the maximum searches, the RSV would be 50.

Seasonality of time series was additionally checked using autocorrelation and partial autocorrelation functions. The keywords were taken from a similar investigation on the topic [22] and translated to Polish, which resulted in 69 words. The data were collected on 14 March 2021 and each keyword was entered separately into Google Trends. The investigated period covered the previous decade (January-2010 to Deccember-2020). The collected data were gathered in an Excel spreadsheet. There were time series with a significant number of zeros (RSV = 0) in different months in the following years. An example is ‘przeczos’, with approximately 78% zeros, which may indicate that searches of this word are rare. We decided to not include 13 time series with more than 10% zeros in further analysis due to the risk of a small number of searches for a given word.

### 2.2. Statistical Analysis

We conducted time series analysis to distinguish seasonal patterns among disease term searches. Data extracted from Google Trends were used without any transformations. Statistical calculations were performed using R 4.0.5 [23]. All time series visualization was produced using the ggplot2 package [24]. We used linear regression to estimate the slope, expressed as changes in RSV per year. To investigate significant secular trends in time series data, Seasonal Mann–Kendal Tests were performed. We extracted the seasonal components using the Classical Seasonal Decomposition by Moving Averages. To determine significant seasonal periods, we fitted TBATS (exponential smoothing state space model with Box–Cox transformation, ARMA errors, trend and seasonal components) models from the forecast package [25].

## 3. Results

We conducted an analysis of the most common dermatology-related search terms in the past ten years on Google (Table 1). This led us to identify the seasonal components in the search volumes for several diseases/symptoms, which are graphically shown in Figure 1. The patient might search Google in two periods: before consulting a primary care physician/dermatologist and after such consultation. It seems reasonable to assume that the more medically sophisticated the search term is, the higher the probability that it was searched for after seeing a doctor. We may consider such searching as indicative of patients’ lack of understanding of information provided by doctors, or their reluctance to ask specific questions during the appointment. In such cases, our results may highlight diseases/symptoms that should be explained more carefully in the given periods.

The results for all time series analyses have been summarized in Table 1. The RSVs of atopic dermatitis (AD), alopecia, chlamydia, freckle, gonorrhea, juvenile acne, pimples, psoriasis, rubella, scabies, skin cancer, syphilis and tinea pedis show statistically significant (*p*-value less than or equal to 0.05) decreasing trends. Annual seasonal variations are evident across almost all disease/symptom searches except alopecia, furuncle, impetigo, lupus, nodulus, osteoarthritis, papule, pimple, seborrheic dermatitis, squama, ulcus and vesicula.

To present the dynamics of disease topic searches with annual seasonality, we divided their seasonal component distribution into five equal-sized consecutive subsets (quantile approach). The darkest and lightest colors represent the highest and the lowest values of the seasonality component, respectively. A graphical representation of the seasonal components of the RSV time series, which contain annual seasonality (TBATS seasonal period is equal to 12 in Table 1), is shown in Figure 2. A visual inspection of the time series in Figure 1 can support the above conclusions as well.

To analyze the seasonality, we must define seasons that occur in Poland. Comparably to the rest of Europe, four major seasons were identified—spring from March to May, summer from June to August, autumn from September to November and winter from December to February [26].

## 4. Discussion

The search behavior patterns using Google Trends have been studied previously in the context of dermatologically related topics such as atopic dermatitis, psoriasis, chilblains, cutaneous infestations with arthropods, hair loss and human papilloma virus [27,28,29,30,31,32,33,34]. We prepared a table that compares the clinical symptoms of common dermatological disorders, their seasonality based on the previous literature and our research (Table 2).

Admittedly, some dermatological symptoms are common and may appear in dermatoses of different origin. Using Google Trends in our analysis, it is not possible to exclude the overlap of the same symptom appearing in different illnesses. Bearing in mind the different incidence and seasonality of certain conditions, in most cases, it is difficult to connect the search terms and their seasonality in Poland with a specific diagnosis.

However, some patterns can be seen in our study. Interestingly, despite the plateauing of the incidence of atopic dermatitis in the previous few years in Europe and North America, our results have shown a significant decreasing trend in both (AD) searches, but not in atopic dermatitis [35]. This may be due to better education on skin lesions as well as on the progression of the atopic condition (many educational campaigns, presentations, lectures online and on TV in previous years). Moreover, some patients may not know the abbreviation for atopic dermatitis, which is AD. Interestingly, between 2016 and 2019, an uptrend in searches regarding atopic dermatitis was noticed in Germany [34]. The evident seasonality of searches, with an increased rate during colder months, may suggest that the needs of patients with atopic dermatitis are unmet during periods with lower temperatures and ultraviolet (UV) radiation levels [34]. Unfortunately, a downtrend has been shown for sexually transmitted diseases (STDs) (chlamydia, gonorrhea and syphilis). An uptrend in the rate of this type of infection was observed up to 2019, inverted significantly in 2020, probably due to the lockdown in Poland [36]. The data that we present, showing decreased searches despite growing infections, may suggest poor sexual education in Poland, which leads to an increased incidence of unprotected sex and, thus, STDs.

Data concerning seasonal trends are also worth discussing. Dermatoses such as atopic dermatitis, psoriasis, seborrheic dermatitis and skin cancers are influenced by UV radiation levels [37]. Terms that are related to atopic dermatitis (atopic dermatitis, eczema), to seborrheic dermatitis (itchiness, dandruff) and to psoriasis (psoriasis) were significantly more often searched for in the winter (January, February). Itchiness may be seen in numerous dermatological disorders and is not a term related only to seborrheic dermatitis. Moreover, this symptom is often confusing to patients. Ultraviolet radiation is a known treatment method for atopic dermatitis and psoriasis, and low exposure to UV radiation is one of the factors that might explain why these diseases may exacerbate during the autumn/winter season. Moreover, searches for UV-exposure-related conditions (melanoma, skin cancer and sunburn) were significantly higher during summer or pre-summer months (May, June). This may be also due to the increased attention that people pay to their bodies in the context of being judged, e.g., on the beach, wearing a swimsuit. For some, it may be a motivation to check if their nevi might indicate cancer lesions.

Fungal infections have also been shown to occur more often in the warmer period of the year [38,39,40]. We have shown that terms related to the most common fungal infections (onychomycosis and tinea pedis) were searched for in May and July accordingly, which could be related to a desire to wear sandals that reveal the feet. However, no seasonal component has been detected in candida searches.

An interesting aspect of search behavior has been observed in the context of STDs, which showed more searches in times of the year opposite to the actual seasonality of the infections. Despite the fact that in most regions of the world, STDs are noticed more often during the summer period, the searches in Poland were significantly higher during winter months (January, February) [41,42,43,44]. One of the explanations that we propose is the increased sex tourism ratio, which may intensify in this period due to the holiday and winter break, when more people decide to travel than normally. A recent article seems to support this thesis, as a longer duration of travel, travelling with friends and being younger have been found to be some of the factors associated with higher levels of casual sex [45].

Blisters, papules and dandruff, as well as asthma, were significantly more often searched for in the period from March to December 2020 in comparison with the same periods in the previous nine years. The year 2020 was the pandemic period in Poland. Different cutaneous manifestations of SARS-CoV-2 (COVID-19) were reported and reviewed previously [46,47]. These mainly include urticaria, maculopapular/morbilliform rash, papulovesicular exanthem, acral chilblain-like pattern, purpura and livedo reticularis [46,47]. In fact, in our study, we have shown that blisters and papules were more often searched for. These terms may be related to the cutaneous SARS-CoV-2 symptoms; however, they are not specific and may be related to a wide variety of dermatological disorders, such as viral infections, dermatitis, impetigo, insect bites, trauma or bullous diseases (pemphigoid, epidermolysis bullosa). Moreover, none of the searches for erythema, petechiae or vesicula demonstrated such search trends. Thus, our data are not sufficient to report any significant differences in the search trends of dermatological terms related to cutaneous COVID-19 symptoms. In fact, only a COVID-19 infection may have been confused with an asthma flare, which resulted in an increased rate of searches.

Our study had some limitations. The sociodemographic data on the people who conducted the searches were not available. The Google Trends tool does not provide absolute quantitative data, only relative percentage data, so it cannot be determined how frequently a given phrase is searched for [48]. Moreover, other studies indicate that women and young people are more likely to seek health-related information online, so searches are not specific for every age and gender group [49,50]. The Google searches do not indicate the actual incidence of symptoms—recommendations are not only influenced by what others searched for [51], but elements of panic and “social contagion” can be present in search behavior [52].

## 5. Conclusions

Despite the limitations that we have described, our study is one of the first to analyze Google search trends for dermatology-related terminology. There is strong potential in the analysis of big data on Internet searches, which, in the future, might be used to enhance the understanding of patients’ needs by doctors and other health professionals [107]. In some cases, this type of study demonstrates a need for a more comprehensive approach to managing patients with certain diseases (e.g., rosacea) [84].

Determining trends is only part of the data needed to try to predict cases. This subject requires further research, and our work is intended to help guide the initial direction of such investigation. Moreover, our article can help dermatologists and medical students who intuitively know which symptoms are seasonal but need a collection of the symptom base in one place, confirming that such a relationship can be used in other studies.

## Figures and Tables

**Figure 1 ijerph-19-08934-f001:**
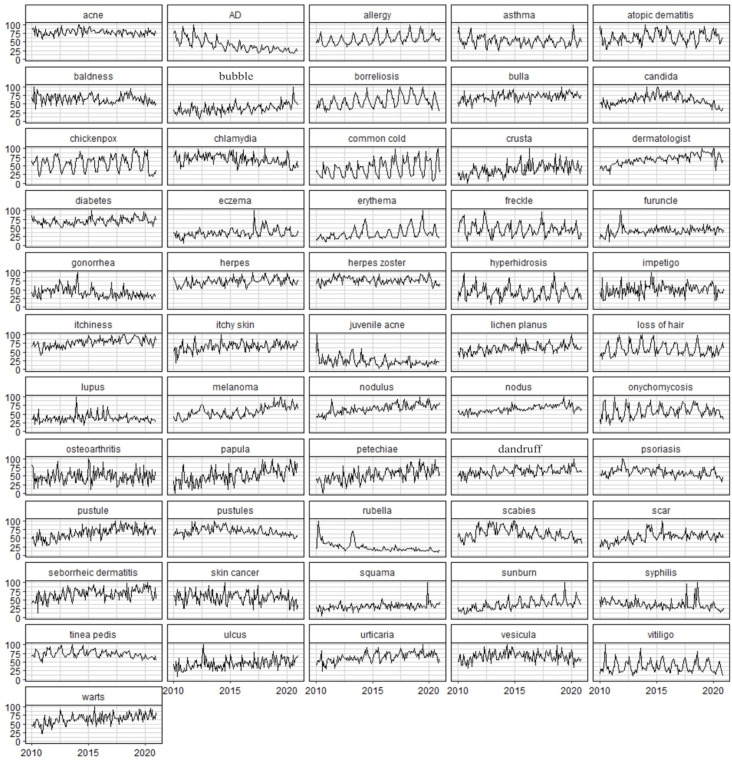
Time series plots for relative search volumes on dermatological topics. The horizontal axis is the date (January-2010–December-2020) and the vertical axis is the value of relative search volume.

**Figure 2 ijerph-19-08934-f002:**
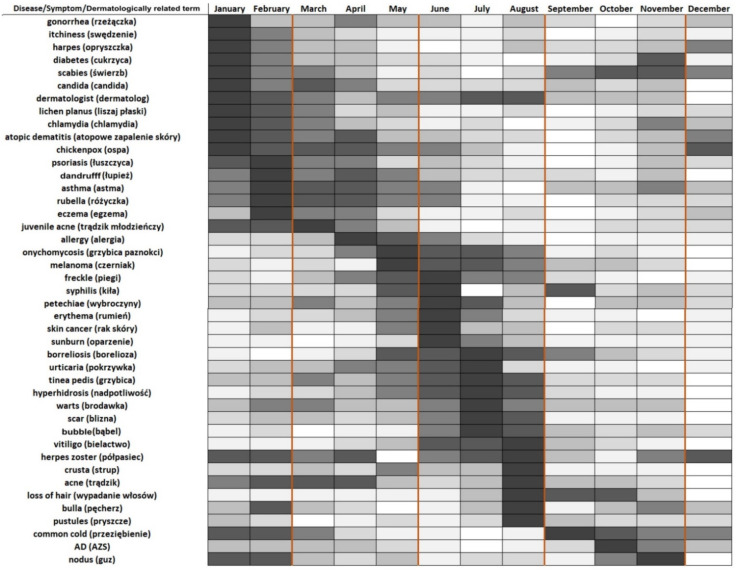
Calendar representation of seasonal components of relative search volumes on dermatological diseases/symptoms. The darkest color corresponds to the highest value of the seasonal component (the month when symptoms are most searched for) and the lightest color corresponds to the lowest value of the seasonal component. Vertical orange lines indicate the division of months into seasons: spring (March, April, May), summer (June, July, August), autumn (September, October, November, December), winter (December, January, February).

**Table 1 ijerph-19-08934-t001:** Time series analysis of dermatologic disease/symptom searches in Google.

Disease/Symptom/Dermatologically Related Term (Polish Translation—Searched Term)	Slope [RSV/Year]	Seasonal Mann–Kendal Test	TBATS Seasonal Periods	Month with the Highest Seasonal Component [RSV]	Month with the Lowest Seasonal Component [RSV]
acne (trądzik)	−0.2	tau = −0.03	12	August [7.22]	December [−10.68]
allergy (alergia)	1.59 ***	tau = 0.62 ***	12	April [22.37]	September [−13.39]
asthma (astma)	−0.66	tau = −0.13 *	12	February [12.65]	August [−20.47]
atopic dematitis (atopowe zapalenie skóry)	0.73	tau = 0.14 *	12	January [18.01]	September [−21.12]
AD (AZS)	−4.15 ***	tau = −0.85 ***	12	October [19.81]	July [−15.63]
alopecia hair loss (łysienie)	−0.7 *	tau = −0.11	NO	-	-
blister bubble (bąbel)	2.09 ***	tau = 0.38 ***	12	July [7.89]	March [−4.91]
borreliosis Lyme disease (borelioza)	1.9 ***	tau = 0.42 ***	12	July [21.68]	February [−17.83]
bulla blister (pęcherz)	1.5 ***	tau = 0.33 ***	12	August [9.06]	May [−7.33]
candida (candida)	−0.46	tau = −0.05	12	January [11.55]	December [−11.10]
chickenpox (ospa)	0.04	tau = 0.24 ***	12	January [18.64]	September [−28.88]
chlamydia (chlamydia)	−1.72 ***	tau = −0.31 ***	12	January [12.30]	August [−9.85]
common cold (przeziębienie)	2.23 ***	tau = 0.48 ***	12	September [24.53]	July [−27.00]
crusta scab (strup)	2.68 ***	tau = 0.44 ***	12	August [16.57]	December [−9.42]
dermatologist (dermatology)	3.32 ***	tau = 0.76 ***	12	January [7.68]	December [−14.22]
diabetes (cukrzyca)	1.05 ***	tau = 0.38 ***	12	January [13.00]	August [−8.70]
eczema (egzema)	1.17 ***	tau = 0.21 **	12	February [13.88]	September [−8.83]
erythema (rumień)	1.9 ***	tau = 0.50 ***	12	June [21.93]	November [−12.18]
freckle (piegi)	−1.06 *	tau = −0.14 *	12	June [22.46]	November [−17.53]
furuncle (czyrak)	0.9 **	tau = 0.27 ***	NO	-	-
gonorrhea (rzeżączka)	−1.72 ***	tau = −0.30 ***	12	January [9.08]	October [−7.34]
herpes (opryszczka)	1.13 ***	tau = 0.33 ***	12	January [14.15]	June [−10.39]
urticaria hives (pokrzywka)	2.33 ***	tau = 0.50 ***	12	July [11.27]	November [−9.29]
hyperhidrosis excessive sweating (nadpotliwość)	−0.71	tau = −0.16 *	12	July [22.38]	December [−17.59]
impetigo (liszajec)	0.68	tau = 0.10	NO	-	-
itchiness (swędzenie)	2.2 ***	tau = 0.50 ***	12	January [12.08]	September [−5.97]
pruritus (świąd)	1.29 ***	tau = 0.22 **	NO	-	-
juvenile acne (trądzik młodzieńczy)	−1.55 ***	tau = −0.24 ***	12	March [11.36]	July [−8.82]
lichen planus (liszaj)	1.86 ***	tau = 0.41 ***	12	January [11.90]	September [−6.97]
loss of hair (wypadanie włosów)	−0.23	tau = −0.11	12	August [24.64]	December [−14.50]
lupus (toczeń)	−0.07	tau = 0.00	NO	-	-
melanoma (czerniak)	3.27 ***	tau = 0.50 ***	12	May [9.17]	December [−7.62]
nodulus lump, small (guzek)	2.96 ***	tau = 0.58 ***	NO	-	-
nodus, lump, large (guz)	2.2 ***	tau = 0.61 ***	12	November [4.70]	December [−4.36]
onychomycosis fungal nail infection (grzybica paznokci)	0.95 *	tau = 0.16 *	12	May [17.88]	December [−16.45]
osteoarthritis (zwyrodnienie stawów)	0.3	tau = 0.01	NO	-	-
papule (grudka)	3.73 ***	tau = 0.43 ***	NO	-	-
petechia (wybroczyny)	3.51 ***	tau = 0.44 ***	12	June [10.63]	September [−14.58]
pustule (krosta)	3.45 ***	tau = 0.50 ***	NO	-	-
pustules (krosty)	−1.12 ***	tau = −0.31 ***	12	August [12.92]	March [−5.70]
dandruff (łupież)	1.51 ***	tau = 0.41 ***	12	February [11.15]	December [−10.92]
psoriasis (łuszczyca)	−1.17 ***	tau = −0.20 **	12	February [10.99]	September [−8.96]
rubella (różyczka)	−2.7 ***	tau = −0.65 ***	12	February [5.04]	September [−4.77]
scabies (świerzb)	−1.36 **	tau = −0.25 ***	12	January [13.69]	July [−17.46]
scar (blizna)	2.01 ***	tau = 0.40 ***	12	July [10.67]	December [−7.64]
seborrheic dermatitis (łojotokowe zapalenie skóry)	2.22 ***	tau = 0.29 ***	NO	-	-
herpes zoster shingles (półpasiec)	−0.22	tau = −0.08	12	August [3.89]	May [−8.05]
skin cancer (rak skóry)	−0.97 *	tau = −0.16 *	12	June [18.75]	September [−9.83]
squama (łuska)	1.18 ***	tau = 0.28 ***	NO	-	-
sunburn (oparzenie)	2.65 ***	tau = 0.62 ***	12	June [20.76]	March [−7.91]
syphilis (kiła)	−1.14 **	tau = −0.33 ***	12	June [7.39]	July [−7.18]
tinea pedis fungal foot infection, athlete’s foot (grzybica stóp)	−0.88 **	tau = −0.23 ***	12	July [11.06]	December [−12.59]
ulcus ulcers (owrzodzenie)	1.75 ***	tau = 0.30 ***	NO	-	-
vesicula (pęcherzyk)	0.06	tau = −0.01	NO	-	-
vitiligo (bielactwo)	−0.28	tau = −0.03	12	August [24.77]	December [−14.97]
wart (brodawka)	2.35 ***	tau = 0.46 ***	12	July [11.99]	December [−14.81]

Note: *, ** and *** denote statistical significance at the 0.05, 0.01 and 0.001 levels.

**Table 2 ijerph-19-08934-t002:** Comparison of the clinical symptoms of the common dermatoses worldwide, their seasonality and related searched terms in Polish Google Trends.

Common Dermatological Diseases Worldwide *	Clinical Symptoms of Common Dermatoses	Seasonality of Common Dermatological Diseases	Related Terms Searched in Google Trends in Poland	Significant Seasonality of Searched Terms in Poland (Month with the Highest Seasonal Component) ^
Acne vulgaris [53] (4.6–85%) **	Papule, pustule, nodulus, scarring [54]	Various results of the studies (an exacerbation during summer and/or winter months is prominent), difficult to reach a consensus [55,56,57,58,59,60,61,62,63,64,65]	acne (trądzik)	No
			juvenile acne (trądzik młodzieńczy)	*p* < 0.001 (March)
			nodulus (guzek)	No
			nodus (guz)	*p* < 0.001 (November)
			papule (grudka)	No
			pustule (krosta)	No
			pustules (krosty)	*p* < 0.001 (August)
			scar (blizna)	*p* < 0.001 (July)
Androgenic alopecia [66,67] (80% in men) (42% in women) ** and other types of alopecia [68] (2%) ** ^§^	Loss of hair, scarring [69]	Seasonality of the types of alopecia—a predilection for colder months [70] Seasonality from Google Trends—significantly correlated, most prevalent in summer and fall [33]	alopecia (łysienie)	No
			crusta (strup)	*p* < 0.001 (August)
			loss of hair (wypadanie włosów)	No
			lupus (toczeń)	No
			scar (blizna)	*p* < 0.001 (July)
Atopic dermatitis [71,72,73,74] (2–10%) **	Crusted scales, eczema, pruritus, dry skin, lichenification [75]	Disease is more severe in winter (more admissions) [40,65]	allergy (alergia)	*p* < 0.001 (April)
			asthma (astma)	*p* < 0.05 (February)
			atopic dematitis (atopowe zapalenie skóry)	*p* < 0.05 (January)
			AD (AZS)	*p* < 0.001 (October)
			crusta (strup)	*p* < 0.001 (August)
			eczema (egzema)	*p* < 0.01 (February)
			erythema (rumień)	*p* < 0.001 (June)
			itchiness (swędzenie)	*p* < 0.001 (January)
			pruritus (świąd)	No
Onychomycosis/Tinea pedis [76] (up to 70% percent in some populations) *	Onychomycosis—hyperkeratinization, change in color of the plate to brown/yellow, onychodystrophy [76] Tinea pedis maceration, exfoliation of the dermis, inflammatory erythema, cracks and oozing erosions [76]	Admissions more often in non-winter months [38,39,40]	candida (candida)	No
			erythema (rumień)	*p* < 0.001 (June)
			itchiness (swędzenie)	*p* < 0.001 (January)
			pruritus (świąd)	No
			loss of hair (wypadanie włosów)	No
			hyperhidrosis (nadpotliwość)	*p* < 0.05 (July)
			onychomycosis (grzybica paznokci)	*p* < 0.05 (May)
			tinea pedis (grzybica stóp)	*p* < 0.001 (July)
Psoriasis [77] (0.51–11.43% in adults, 0–1.37% in children) *	Well-defined, sharply demarcated erythematous plaques, erythrodermic, pustules, guttate, psoriatic onychodystrophy, psoriatic arthritis [78]	Seasonal variability in approximately 50% patients, some studies report significantly more admissions during winter [37,40,59,79,80]	erythema (rumień)	*p* < 0.001 (June)
			papule (grudka)	No
			pustule (krosta)	No
			pustules (pryszcze)	*p* < 0.001 (August)
			psoriasis (łuszczyca)	*p* < 0.01 (February)
			squama (łuska)	No
Rosacea [81,82] (2.1–10%) **	Frequent flushing, persistent erythema and telangiectasia, inflammation with swelling, papules, pustules [83]	May be aggravated in summer [84]	eczema (egzema)	*p* < 0.01 (February)
			erythema (rumień)	*p* < 0.001 (June)
			lupus (toczeń)	No
Skin cancer (NMSC + melanoma) [85,86] (1/100 000–1000/100 000 for NMSC, 21.8/100 000 for melanoma) ***	NMSC—nodule (erythematous, keratinizing, crusted), hyperkeratosis, ulceration, scarring [87,88] Melanoma—asymmetrical, fast-growing macule (superficial type), nodule/nodus (nodular type) [89]	NMSC may be more commonly detected in summer [37]. UV exposure (main risk factor) has the most significant impact during this season [90]	crusta (strup)	*p* < 0.001 (August)
			melanoma (czerniak)	*p* < 0.001 (May)
			nodulus (guzek)	No
			nodus (guz)	*p* < 0.001 (November)
			papule (grudka)	No
			scar (blizna)	*p* < 0.001 (July)
			skin cancer (rak skóry)	*p* < 0.05 (June)
			sunburn (oparzenie)	*p* < 0.05 (June)
Staphylococcus aureus infections [91,92,93] (Impetigo—8.4–19.4%, Furunculosis—unknown) **	A thin-walled vesicle -> rapidly ruptures -> superficial erosion covered with yellowish-brown or honey-colored crusts. The crusts may dry, separate and disappear, leaving a red mark that heals without scarring [94]	More common in summer and autumn [40,95]	blister (bąbel)	*p* < 0.001 (July)
			bulla (pęcherz)	*p* < 0.001 (August)
			crusta (strup)	*p* < 0.001 (August)
			diabetes (cukrzyca)	*p* < 0.001 (January)
			furuncle (czyrak)	No
			impetigo (liszajec)	No
			squama (łuska)	No
			ulcus (owrzodzenie)	No
			vesicula (pęcherzyk)	No
Seborrheic dermatitis [96,97] (1–3% in adults, up to 42% in children) **	Skin flakes (dandruff) on scalp, hair, eyebrows, beard or mustache Patches of greasy skin covered with flaky white or yellow scales or crust on the scalp, face, sides of the nose, eyebrows, ears, eyelids, chest, armpits, groin area or under the breasts, erythema, itching [98]	More common in winter, but not according to all studies [37,65,79,99]	crusta (strup)	*p* < 0.001 (August)
			erythema (rumień)	*p* < 0.001 (June)
			itchiness (swędzenie)	*p* < 0.001 (January)
			dandruff (łupież)	*p* < 0.001 (February)
			pruritus (świąd)	No
			seborrheic dermatitis (łojotokowe zapalenie skóry)	No
			squama (łuska)	No
Sexually transmitted diseases (STDs) [100,101] (374 million worldwide per year) ***	Mostly asymptomatic, vaginal discharge, urethral discharge or burning in men, genital ulcers and abdominal pain [100,101]	More common in summer [41,42,43,44]	chlamydia (chlamydia)	*p* < 0.001 (January)
			gonorrhea (rzeżączka)	*p* < 0.001 (January)
			itchiness (swędzenie)	*p* < 0.001 (January)
			pruritus (świąd)	No
			scabies (świerzb)	*p* < 0.001 (January)
			syphilis (kiła)	*p* < 0.001 (June)
			warts (brodawki)	*p* < 0.001 (July)
Viral infections [102,103,104](HPV—9–13%,HSV1—66%,HSV2—13.2% (95% Confidence Interval 11.55—16.39%)) **	HPV—genital warts—flat lesions, small cauliflower-like bumps or tiny stem-like protrusions, common warts appear as rough, raised bumps and usually occur on the hands and fingers [105] HSV—pain or itching, small red bumps or tiny white blisters. These may appear a few days to a few weeks after infection. Ulceration, small crust (scabs) [106]	We did not manage to find any data	chickenpox (ospa)	*p* < 0.001 (January)
			herpes (opryszczka)	*p* < 0.001 (January)
			rubella (różyczka)	*p* < 0.001 (February)
			herpes zoster (półpasiec)	No
			vesicula (pęcherzyk)	No
			warts (brodawka)	*p* < 0.001 (July)

* According to prevalence, prevalence rates and incidence rates. ** Prevalence rates. *** Incidence rates. ^§^ Excluding androgenic alopecia. ^ Measured with Seasonal Mann–Kendal Test. Searches that were not related to the most common diseases—borreliosis (bolerioza), common cold (przeziębienie), dermatologist (dermatolog), freckle (piegi), urticaria (pokrzywka), lichen planus (liszaj), osteoarthritis (zwyrodnienie stawów), petechia (wybroczyny), vitiligo (bielactwo).

## Data Availability

Data without statistical analysis available on Google Trends.

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
