# Peer review of "Seasonal Patterns and Trends in Dermatoses in Poland"

_ijerph, 2022, doi:10.3390/ijerph19158934_

Round 1
Reviewer 1 Report
This proposal is actually a good piece. Though the methodology used here is appropriate, it would have been desirable to mention may be in 1 or 2 sentences, the existence of a close method (systematic reviews & meta-analysis), and then state why the choice of Google trends-relative search volume (RSV) was the best.
On a different point, I wonder about the real scope or real meaning of this study. The author must not make inferences between internet search and actual morbidity. Individuals can search on a disease symptom not because they are sick, but for other purposes such as scientific curiosity or academic reason. That said, I view this work as a simple scientific exercise, not as practical research that is whose results can serve as tools for decision-making.
Author Response
Thank you very much for your revision and kind words.
Google Trends was chosen because Google is the most popular search engine in the world and Poland, as indicated in the manuscript. It is valid that trends alone should not be used to make decisions and take action. This has been demonstrated with influenza prediction trials (Lazer, David, et al., "The parable of Google Flu: traps in big data analysis." Science 343.6176 (2014): 1203-1205.). Determining trends is only part of the data needed to try to predict cases - incl. the specific time of the day of search and the frequency with which the user does it. This subject requires further research, and our work is intended to help guide the initial direction of such investigation.
Moreover, our article can help dermatologists who intuitively know what symptoms are seasonal. However, we realized we needed studies such as the one that would collect the symptom base in one place and confirm that such a relationship can be used in other studies.
Reviewer 2 Report
Please see attachment

Author Response
Dear Authors,
Thank you for the opportunity to peer review your work in researching this interesting theme.
In order to improve your manuscript, please take into consideration the following aspects:
- At lines 74-75 you wrote: “The words whose search frequency (RSV) was 0 in at least two months were not used for further analysis due to low prevalence.” Please offer a more extended explanation of this decision regarding the relevance of length of time period (2 months) keeping in mind the seasonality of the search. Also, please mention how many keywords were excluded from the initial 69 (see also #5).
Response: Thank you for that reasonable suggestion. We have developed the description with the percentage of zeros in the entire time series, as a risk of data imprecision (very small number of searches).
- At lines 94-95 and 96-97 the phrase (“We may consider such searching as indicative of patients’ lack of understanding of information provided by doctors, or their reluctance to ask specific questions during the appointment.”) repeated itself.
Response:Thank you for this comment.
- At line 102 and 117 please renumber Figure 2 as Figure 1 and vice-versa
Response:The numbering of figures has been corrected.
- At line 105 please include the full word before using the acronym AD.
Response: We have included the full phrase atopic dermatitis before the acronym.
- Please explain why at lines 100-101 (in figure 2) there are 56 illnesses presented and at lines 116-117 (in figure 1) there are only 43, while initially there were 69 keywords selected (see #1 again).
Response: We have included information about constraint (time series has to have seasonality to visualize it) based on Table 1.
- At lines 110-114 please explain how the approach of dividing data into five equal intervals matches the visual representations of the seasonality with 4 seasons.
Response: Five different intervals based on the quantile approach (each time series has a different range of seasonal components values) have only been used as a method of visualizing changes in seasonal components. We chose to split the data into five equal intervals as they best visualize changes over the year. It is not connected with the number of seasons.
- Please note that ‘dermatologist’ is neither a disease nor a dermatological symptom (see the headline in figure 1 and table 1).
Response: We have corrected the headlines of figure 1 and table 1 by adding “dermatologically related terms” so that all of the searches are suitable.
- At line 129, the bibliographical reference should refer to Poland rather than Germany.
Response: The climate in both countries is very similar, and we found this work the most accurate.
- In table 1 please explain why ‘pustule’ and pustules’ ought to be considered as separate keywords.
Response: It was singular in the study from which we took the word base. Our team subjectively felt that the plural is more often used when talking about this symptom, so we thought it was worth including.
- At line 268 please insert the full names of the authors for the reference [22].
Response:Thank you for your comment - the citation has been corrected.
11. For broadening your audience, in the Discussion paragraph please refer also to the following aspects:
- How the students in Medicine can benefit from the study results?
- How the public health policy makers can benefit from the study results?
- What are the implications of seasonality of the mentioned diseases for physicians, pharmacists and drug manufacturers?
Response: Thank you for that suggestion. A paragraph has been added to the conclusion on this subject. Although this study should be seen more as a summary and direction, it does not exhaust the topic. The results presented here without much context may not be sufficient for public health policymakers. However, the methodology stands out from other research from Google Trends, and medical students will be able to use it in their investigations.
Reviewer 3 Report
The authors present an interesting work using a novel tool, Google trends, which can be very useful in exploring patient behaviour regarding different diseases using real-time data. In their study, they address the seasonability of different research terms related to several dermatoses in Poland. The methodology is correct and the figures and tables are clear but it has several major drawbacks. First of all, the relevance of the study is not very clear. It is hard to understand the knowledge gap they are trying to fill. Also, even though some interesting conclusions about patients’ behaviour are presented, they are not always accurately supported by the study’s results.
Regarding RSV, I think there seems to be some confusion. It is stated that ". Periods when the given word was not searched at all mean RSV = 0, and the peak of popularity defines RSV = 100. Likewise, when a given term was searched with 50% frequency of the maximum searches, the RSV would be 50." I believe this is inaccurate. RSV measures the popularity of a research term when compared to other Google terms. 0 does not mean that a word was not searched at all, only that it has been searched in a small number compared to peaks.
Finally, there are some minor corrections that should be made:
1. Figure 2 is mentioned before figure 1.
2. Line 143. Other causes of decreased searches for atopic dermatitis (including media coverage) should be mentioned.
3. Line 153. It is stated that itchiness is a term which is only related to seborrheic dermatitis. This seems very inaccurate. It might be a misunderstanding but rephrasing is certainly needed.
4. Lines 185-188. Again, I believe media coverage of Covid19 has probably greatly influenced Covid related searches. Besides, blisters and papules are very unspecific terms that may be related to a wide array of dermatological conditions. Their relation with Covid19 cutaneous manifestations seems very speculative.
Author Response
The authors present an interesting work using a novel tool, Google trends, which can be very useful in exploring patient behaviour regarding different diseases using real-time data. In their study, they address the seasonability of different research terms related to several dermatoses in Poland. The methodology is correct and the figures and tables are clear but it has several major drawbacks. First of all, the relevance of the study is not very clear. It is hard to understand the knowledge gap they are trying to fill. Also, even though some interesting conclusions about patients’ behaviour are presented, they are not always accurately supported by the study’s results.
Response: Thank you for your revision.
All comments regarding inaccuracies based on the obtained results have been included in this review and in other reviewers.
Determining trends is only part of the data needed to try to predict cases - incl. the specific time of the day of search and the frequency with which the user does it. This subject requires further research, and our work is intended to help guide the initial direction of such investigation.
Moreover, our article can help dermatologists who intuitively know what symptoms are seasonal. However, we realized we needed studies such as the one that would collect the symptom base in one place and confirm that such a relationship can be used in other studies.
Regarding RSV, I think there seems to be some confusion. It is stated that ". Periods when the given word was not searched at all mean RSV = 0, and the peak of popularity defines RSV = 100. Likewise, when a given term was searched with 50% frequency of the maximum searches, the RSV would be 50." I believe this is inaccurate. RSV measures the popularity of a research term when compared to other Google terms. 0 does not mean that a word was not searched at all, only that it has been searched in a small number compared to peaks.
Response: We fully agree, RSV = 0 may or may not be 0 for the lookup number. We have changed this part of the text.
Finally, there are some minor corrections that should be made:
- Figure 2 is mentioned before figure 1.
Response:The numbering of figures has been corrected.
- Line 143. Other causes of decreased searches for atopic dermatitis (including media coverage) should be mentioned.
Response: We have mentioned the possible causes of decreased searches and contrasted this with a paper, in which between 2016-2019 there has been an increased volume of searches in Germany.
- Line 153. It is stated that itchiness is a term which is only related to seborrheic dermatitis. This seems very inaccurate. It might be a misunderstanding but rephrasing is certainly needed.
Response: We have rephrased this line by cuting the word “only”, which may have been confusing. Moreover, we have provided an explanation to the usage of term “itchiness”, which is seen in numerous dermatological disorders and may be confused by patients.
- Lines 185-188. Again, I believe media coverage of Covid19 has probably greatly influenced Covid related searches. Besides, blisters and papules are very unspecific terms that may be related to a wide array of dermatological conditions. Their relation with Covid19 cutaneous manifestations seems very speculative.
Response: We have clarified, that these symptoms may be a sign of COVID19 manifestation, however, thay may also be seen in other dermatological disorders.
Reviewer 4 Report
The authors present an interesting paper on Google trends in relation to dermatological conditions in Poland.
My major concern is the selection of words included in the search. The authors rely on previous work, but why do they choose these words and not others? Common lesions or pathologies, which can lead to a large number of searches, e.g. nevus, actinic keratoses, seborrhoeic keratoses or hidradenitis suppurativa, to name but a few, seem to have been left out of the study (at least not in the figures and tables). The authors should detail which words have been included and justify why they have included those and not others, as this may lead to a significant lack of information.
Other minor comments:
- Please specify abbreviations used in figures (AD = atopic dermatitis?).
- Please include all tables in the results section.
- Authors should include in the discussion other important work carried out on Google trends in dermatology:
Pereira MP, Ziehfreund S, Rueth M, Ewering T, Legat FJ, Lambert J, Elberling J, Misery L, Brenaut E, Papadavid E, Garcovich S, Evers AWM, Halvorsen JA, Szepietowski JC, Reich A, Gonçalo M, Lvov A, Bobko S, Serra-Baldrich E, Wallengren J, Savk E, Leslie T, Ständer S, Zink A. Google search trends for itch in Europe: a retrospective longitudinal study. J Eur Acad Dermatol Venereol. 2021 Jun;35(6):1362-1370. doi: 10.1111/jdv.17077.
Mick A, Tizek L, Schielein M, Zink A. Can crowdsourced data help to optimize atopic dermatitis treatment? Comparing web search data and environmental data in Germany. J Eur Acad Dermatol Venereol. 2022 Apr;36(4):557-565. doi: 10.1111/jdv.
Martinez-Lopez A, Ruiz-Villaverde R, Molina-Leyva A. Google search trends in psoriasis: a pilot evaluation of global population interests. J Eur Acad Dermatol Venereol. 2018 Oct;32(10):e370-e372. doi: 10.1111/jdv.14944.
Seasonality of hair loss: a time series analysis of Google Trends data 2004-2016. Br J Dermatol. 2018 Apr;178(4):978-979. doi: 10.1111/bjd.16075.
Author Response
My major concern is the selection of words included in the search. The authors rely on previous work, but why do they choose these words and not others? Common lesions or pathologies, which can lead to a large number of searches, e.g. nevus, actinic keratoses, seborrhoeic keratoses or hidradenitis suppurativa, to name but a few, seem to have been left out of the study (at least not in the figures and tables). The authors should detail which words have been included and justify why they have included those and not others, as this may lead to a significant lack of information.
Response:Thank you for that suggestion. The main goal of our study was to investigate the seasonality of the most common skin-related keywords. All the words were taken from the article we were based on, and we found this word base sufficient. Words that had RSV = 0 are available in the previous article and are easily accessible to the reader. They were not included in the work as they were not analyzed further.
Other minor comments:
- Please specify abbreviations used in figures (AD = atopic dermatitis?).
Response: We have specified the abbreviation before using it in the text.
- Please include all tables in the results section.
Response: All tables have been included in the results section.
- Authors should include in the discussion other important work carried out on Google trends in dermatology:
Pereira MP, Ziehfreund S, Rueth M, Ewering T, Legat FJ, Lambert J, Elberling J, Misery L, Brenaut E, Papadavid E, Garcovich S, Evers AWM, Halvorsen JA, Szepietowski JC, Reich A, Gonçalo M, Lvov A, Bobko S, Serra-Baldrich E, Wallengren J, Savk E, Leslie T, Ständer S, Zink A. Google search trends for itch in Europe: a retrospective longitudinal study. J Eur Acad Dermatol Venereol. 2021 Jun;35(6):1362-1370. doi: 10.1111/jdv.17077.
Mick A, Tizek L, Schielein M, Zink A. Can crowdsourced data help to optimize atopic dermatitis treatment? Comparing web search data and environmental data in Germany. J Eur Acad Dermatol Venereol. 2022 Apr;36(4):557-565. doi: 10.1111/jdv.
Martinez-Lopez A, Ruiz-Villaverde R, Molina-Leyva A. Google search trends in psoriasis: a pilot evaluation of global population interests. J Eur Acad Dermatol Venereol. 2018 Oct;32(10):e370-e372. doi: 10.1111/jdv.14944.
Hsiang EY, Semenov YR, Aguh C, Kwatra SG. Seasonality of hair loss: a time series analysis of Google Trends data 2004-2016. Br J Dermatol. 2018 Apr;178(4):978-979. doi: 10.1111/bjd.16075.
Response: Thank you for that resonable suggestion. We have included those articles, which we believe have been important in google trends data analysis in dermatology. Moreover, we have performed an additional search to include other important papers from this field in our discussion.
Round 2
Reviewer 3 Report
I believe the authors have taken notice of the comments made by all reviewers and the quality of the paper has been clearly improved. I believe it should be accepted for publication.